# A Simplified Frequency-Domain Feedback Active Noise Control Algorithm

**Yuan Gao [1], Guoliang Yu [1] and Min Gao [2,\*]**

[1] College of Information Engineering, China Jiliang University, Hangzhou 310018, China; 18258886572@163.com (Y.G.); glyu@cjlu.edu.cn (G.Y.)

[2] College of Sciences, China Jiliang University, Hangzhou 310018, China

\* Correspondence: gaomin@cjlu.edu.cn

**Abstract:** When the adaptive filter length is increased, the calculation complexity increases rapidly because the relationship between the calculation and the adaptive filter length $N$ contains a power function with no secondary path identification algorithm. Under the basic premise of unreduced noise reduction, herein, a simplified frequency-domain feedback active noise control algorithm is proposed. To reduce the computation complexity, the total delay is adopted as the estimated secondary path; the filtered reference signal is produced in the frequency domain by using multiplication to replace convolution calculation in the time domain and then updating the adaptive filter coefficients in the frequency domain. Therefore, the computational complexity has a logarithmic function with the increased adaptive filter length in the proposed algorithm. If the adaptive filter length is 512, the existing WSMANC algorithm's calculation is 271,360 real number multiplications, while that of the proposed algorithm is only 38,912 real number multiplications. To verify the proposed algorithm's stability, convergence speed, and noise reduction, the single-frequency noise, narrowband white noise, and narrowband pink noise, respectively, are used as the primary noise types in the simulations. The results show that (1) the proposed SFDFBANC algorithm can obtain similar noise reduction performance to existing algorithm, (2) the convergence rate is faster than existing algorithm, and (3) if the adaptive filter length is more than 64, the proposed algorithm exhibits a lower computational complexity.

**Keywords:** active noise control; no secondary path identification; feedback system; low computation complexity

## 1. Introduction

Excessive noise causes harm to our physical and psychological health. Noise pollution has become one of the four major public hazards today [1]. Generally speaking, passive noise control methods can be adopted to reduce noise, and common methods include sound absorption technology, insulating the noise, etc. [2]. The principle of these methods is that the sound energy is consumed by the influence between the material and sonic waves in [3]. The effect on medium- and high-frequency noise is better than that on low-frequency noise, which remains prone to diffraction due to its long wavelength, limiting the noise reduction effect. Active noise control technology is often used to degrade low-frequency noise [4,5].

The principle of "noise elimination by sound" is adopted in active noise control technology to reduce noise by producing "anti-noise" that matches the noise source in size and has an inverse phase [6]. It is classified into a feedforward control system and feedback control system using different control methods. If the noise source information can be conveniently obtained in an active control system, the feedforward active noise control system can be used, and the controller is completed by the feedforward adaptive filter [7,8]. In the feedforward active control algorithm, the filtered-$x$ least mean square algorithm has wide applications because of its simple structure, easy realization, and good noise

reduction effect [9–11]. When primary noise cannot be acquired, but the expected signal and error signal can be obtained simultaneously through the error sensor, the feedback system is used in active noise control [12,13]. In addition, this method has been successfully applied in some engineering projects, such as active noise reduction earplugs [14,15] and active noise reduction headrests [16,17].

A single-channel feedback active control algorithm was proposed by Erisson and widely used [18]; then, it was improved by Popovich et al. and applied to the multichannel control system [19]. Elliott et al. proposed the delay-less algorithm in the frequency-domain feedback system [20,21]. In the existing algorithms, feedback control is often transformed into feedforward control through an internal synthesis reference signal to achieve feedback active noise control [22]. The internal model structure is usually employed due to its simple mechanism, easy implementation, and wide application. Firstly, adaptive filter output is first prefiltered by the true secondary path; then, it is offset by an error signal to obtain the internal synthesis reference signal. In this algorithm, assuming perfect modeling, the modeling secondary path information is completely consistent with the actual secondary path [23,24]. However, the difference always existed between model secondary path information and the actual secondary path. Moreover, the adaptive filter is the inverse of the model secondary path impulse, and the secondary path is a nonminimum phase system, so perfect noise reduction is difficult to realize in a feedback ANC system. Wu et al. adopted the internal model structure and proposed a simplified feedback control algorithm (abbreviated to SAIMC algorithm). The internal reference signal was directly derived from the error signal detected by error microphone, and then the feasibility and stability of the simplification were analyzed, the feedback control system structure was simplified, and the number of calculations was reduced [25]. However, the secondary path information still needs to be obtained for these algorithms first.

To model the information of secondary path impulse response, the online modeling method [26,27] or offline modeling method [28] can be used. However, errors always existed between the actual and modeled secondary path information. If these errors are large, the system is unstable [29]. Zhou et al. proposed to choose the different phases to estimate the phase response that can select the best update direction for the frequency in the 0° and 180° directions to approach the phase information corresponding to the secondary path impulse response [30]. Wu et al. improved this method and applied it to the frequency domain, adding the two directions of ±90° so that the frequency near ±90° could also have fast convergence [31]. Gao et al. realized the four update directions of 0°, 180°, and ±90° in the time domain using the Hilbert transform and estimated the total acoustic delay to compensate part of the secondary path information, which could reduce the number of subbands when realizing broadband noise active control, but the convergence speed was slow [32]. Chen et al. improved the algorithm, adopted the phase estimation method in each subband, and applied it to the frequency domain to reduce the calculation amount [33,34].

To avoid modeling the secondary path information, Gao et al. proposed the feedback active control algorithm without secondary path modeling (referred to as the WSMANC algorithm) [35]. The algorithm analyzes the secondary path impulse response, which is composed of direct sound delay and a series of reflected sound delays and electrical delays. The direct sound and electrical delay are estimated and added together as the total delay, which can then be used as the estimated secondary path impulse response. Because it avoids modeling the secondary path, the calculation complexity is notably reduced; however, it still shows power function growth with the increase in the adaptive filter length.

Due to updating the coefficients of adaptive controllers in the time domain, the filtered reference signal was obtained by convolving the estimated secondary path with the internally generated reference signal. In addition, the two processes are the main components that affect computational complexity. The relationship between calculation complexity and the adaptive filter length contains a power function. Therefore, the computational

complexity will rapidly increase by increasing the adaptive filter length. To further decrease the calculation complexity, a simplified frequency-domain feedback ANC algorithm is proposed. It adopts the delay-less frequency-domain algorithm structure to update the adaptive filter in the frequency domain, estimate the total delay of the secondary path, and use multiplication to replace the convolution calculation in time domain to produce the filtered reference signal.

The subsequent parts of this paper provide the following information. Section 2 derives the simplified frequency-domain feedback active noise control algorithm. Section 3 demonstrates the simulation results in detail. Section 4 addresses calculation analyzation. Finally, concluding remarks can be found in Section 5.

## 2. The Simplified Frequency-Domain Feedback Active Noise Control Algorithm

### 2.1. Estimated the Total Delay

The secondary path is composed of a sensor, a controller, a primary loudspeaker, a secondary loudspeaker, and a related circuit, as well as the acoustic path between the secondary loudspeaker and the error sensor [6]. Thus, the secondary path impulse response is always composed of electrical delay and acoustic delay components, while the acoustic delay component is composed of a direct acoustic delay and a series of reflected acoustic delays, among which the influence of the reflected acoustic delay is small. However, the electrical delay introduced by the sensor, controller, and related peripheral circuit is negligible and can be disregarded [6]; only the input signal delay from the input to the output end of the controller is considered. As such, the total delay $\tau$ of secondary path can be approximately expressed as the summation of electrical delay $\tau_1$ and direct acoustic delay $\tau_2$, which is denoted as the delay from the secondary loudspeaker to the error sensor [32]

$$\tau = \tau_1 + \tau_2 \tag{1}$$

In Formula (1), $\tau_1 = Q/f_s$, where $f_s$ denotes the sampling frequency. $Q$ denotes the delay number from the controller input end to the output end of the reference signal. In addition, $\tau_2 = d/c$, where $d$ is the distance between the secondary loudspeaker and the error sensor, and $c$ represents the sound velocity, which is generally taken as 340 m/s in air.

The digital controller based on an AD21469 digital signal evaluation board (Analog Devices, Wilmington, MA, USA) was used in the test, and its sampling frequency was 48,000 Hz. The sampling interval is about $2.08 \times 10^{-5}$ s between two adjacent sampling data. With the aim of improving the computational efficiency, the down sampling technical is adopted. After 20 instances of down sampling, the sampling frequency was 2400 Hz. If the sampling rate and the upper cutoff frequency of primary noise do not satisfy the Nyquist sampling theorem, it will induce signal distortion, induce the loss of high-frequency details, and reduce the system character. If it satisfies the Nyquist sampling theorem, these questions can be avoided. As the sampling rate is 2400 Hz, the upper cutoff frequency of primary noise is no more than 1200 Hz to satisfy the Nyquist sampling theorem.

Using white noise as the input signal, the electrical delay $\tau_1$ was tested, and 15 sampling points were measured by shorting the input and the output ends of the controller. Thus, we determined that the electrical delay $\tau_1$ was about 6.25 ms.

### 2.2. The Derivation of the WSMANC Algorithm

Figure 1 shows a physical representation of feedback ANC system. Both the desired signal and error signal can be detected simultaneously using an error microphone. The desired signal comes from the primary noise. Then the error signal is superposed by the desired signal, and the signal is produced by the secondary loudspeaker. Then the error signal is fed back to the adaptive filter to produce the anti-noise using feedback ANC algorithms.

Primary Noise

Secondary
Loudspeaker

Error Microphone

Adaptive Filter

**Figure 1.** Physical block diagram of the feedback ANC system.

Figure 2 displays the principal flow chart of the WSMANC algorithm. $x(n)$ denotes the internal synthesis input signal, $d(n)$ denotes the original interference, $y(n)$ is the output of the adaptive filter, $s(n)$ is detected by the error sensor from the secondary loudspeaker, the error signal is represented as $e(n)$, $x_f(n)$ is the time-delay filtering reference signal, $W(z)$ represents the adaptive controller, $C(z)$ is the true secondary path, and $\Delta$ denotes the total acoustic delay.

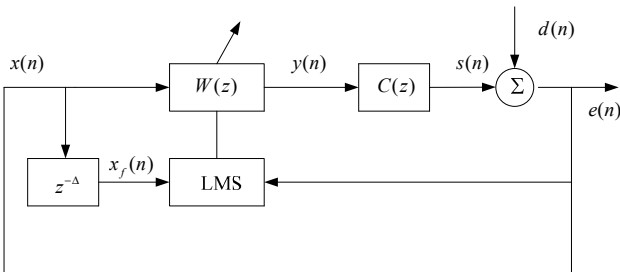

**Figure 2.** Principal flowchart of the WSMANC algorithm.

Based on the SAIMC algorithm, the error signal includes primary noise information; therefore, it can be used as the internal input signal for simplicity The internal synthesized signal $x(n)$ is expressed as follows:

$$x(n) = e(n) \tag{2}$$

The input signal $y(n)$ of the secondary loudspeaker is the internal synthesized reference signal $x(n)$ filtered by the $N$-order adaptive filter $W(z)$ as follows:

$$y(n) = \sum_{i=0}^{N-1} x(n-i)w(i) \tag{3}$$

where $\{w(0)\ldots w(N-1)\}$ are the coefficients of $W(z)$.

The signal $s(n)$ is detected using an error microphone and produced using a filtered output signal $y(n)$ with the secondary path transfer function $C(z)$ as follows:

$$s(n) = \sum_{i=0}^{L_p-1} y(n-i)c(i) \tag{4}$$

where $c(i)$ is the coefficients of the secondary path impulse response, $I = 0, 1, \ldots, L_p - 1$, and $L_p$ denotes the length of the modeling secondary path. For the sake of calculation simplicity, the secondary path length $L_p$ is equal to the adaptive filter length $N$.

$\Delta$ is the sampling number, and it can be calculated using the total delay $\tau$ and the sampling frequency. The time-delay filtering signal $x_f(n)$ is obtained as follows:

$$x_f(n) = x(n - \Delta) \tag{5}$$

The signal $s(n)$ is added to the original interference signal $d(n)$, which produces the error signal $e(n)$ as follows:

$$e(n) = d(n) + \sum_{i=0}^{L_p-1} y(n-i)c(i) \tag{6}$$

The objective function $J(n)$ is written as follows:

$$J(n) = e^2(n) \tag{7}$$

Based on the gradient descent algorithm, the adaptive filter coefficient $w(n)$ is updated in the time domain as follows:

$$w(n) = w(n-1) - \mu_0 x_f^T(n)e(n) \tag{8}$$

where the variable factor $\mu_0$ is a positive value and denotes the step size.

$$w(n) = [w(N-1) \quad w(N-2) \quad \ldots \quad w(0)]^T \tag{9}$$

$$x_f(n) = [x_f(0) \quad x_f(1) \quad \ldots \quad x_f(N-1)]^T \tag{10}$$

The process of converting Equation (2) to (10) above is called the narrowband feedback active noise control algorithm without secondary path modeling algorithm (WSMANC for short).

### 2.3. The Proposed Algorithm

Figure 3 illustrates the configuration block diagram of the SFDFBANC algorithm. Differentiated from Figure 2, the internal synthesis input signal is denoted as $x(n)$, $W(k)$ expresses the frequency-domain adaptive controller, $\hat{C}^*(k)$ can be obtained using the FFT transform of the estimated total delay, $\hat{R}^*(k)$ represents the frequency-domain filtering reference signal, $E(k)$ is the FFT transform of error signal $e(n)$, variable factor $k$ denotes the time exponent in the frequency domain, and the symbol * represents the conjugate.

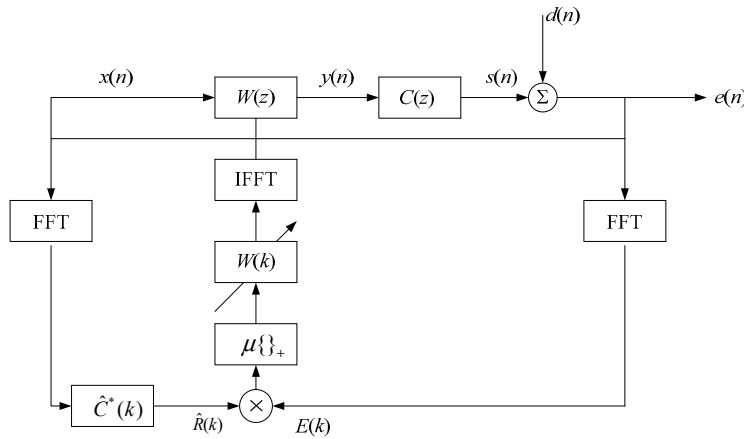

**Figure 3.** Block diagram of SFDFBANC algorithm.

The overlap storage method, which has a 50% overlap rate, is adopted [36]. Fourier transform (FFT) can be utilized by the two consecutive data blocks with length $N$ of the internal synthesized signal to generate the frequency domain signal $X(k)$, which is expressed as follows:

$$X(k) = \text{diag}\{\text{FFT}[x(kN-N) \quad \ldots \quad x(kN) \quad \ldots \quad x(kN+N-1)]\} \tag{11}$$

where $k$ is an integer, and it denotes the frequency-domain index.

$\hat{R}(k)$ is the filtered reference signal in the frequency domain, and it is denoted as follows:

$$\hat{R}(k) = X(k)\hat{C}^*(k) \tag{12}$$

$\hat{C}(k)$ refers to the Fourier transform result of the estimated total direct sound delay $\Delta$ with a length of 2N, with $\Delta$ being the summation of the direct sound delay and the electric delay of circuit [36].

To avoid the "winding" value introduced when calculating linear convolution with cyclic convolution, the frequency-domain error signal $E(k)$ with a length of 2N is expressed as follows:

$$E(k) = \text{FFT}\begin{bmatrix} 0 \\ e(k) \end{bmatrix} \tag{13}$$

where 0 denotes an all-zero column vector of length N, and $e(k)$ can be denoted as follows:

$$e(k) = [e(kN), e(kN + 1), \ldots, e(kN + N - 1)]^{\text{T}} \tag{14}$$

$W(k)$ is the adaptive filter coefficient, and it is updated in the frequency domain as follows:

$$W(k) = W(k - 1) - \mu\hat{R}^*(k)E(k) \tag{15}$$

where variable $\mu$ is a positive value that denotes the step size, $W(k)$ is 2N in length, and $k$ takes the value of 0, 1, . . ., 2N − 1.

The updated formula of the time-domain adaptive filter coefficients $w(n)$ is as follows:

$$w(n) = \text{IFFT}[W(k)]_+ \tag{16}$$

where $w(n)$ is a column vector of length N and "+" denotes the value of IFFT[$W(k)$] from 0 to N − 1.

The derivation process of Formulas (1)–(6) and Formulas (11)–(16) above is called the simplified frequency-domain feedback active noise control algorithm (SFDFBANC).

## 3. Simulation

The sound field environment selected for the simulation experiment is the sound absorption duct sound field, as shown in Figure 4. The duct is a straight square duct with a port diameter of 17 cm, one end of which is closed and the other end of which contains sound-absorbing cotton. The error sensor is situated in the downstream duct about 34 cm away from secondary loudspeaker, and the noise source is in the upstream duct. The distance is about 136 cm from the primary noise source to the secondary loudspeaker. Therefore, the direct acoustic delay $\tau_2$ was about 1 ms, the total delay $\tau$ was about 7.25 ms, and $\Delta$ corresponded to 18 samples.

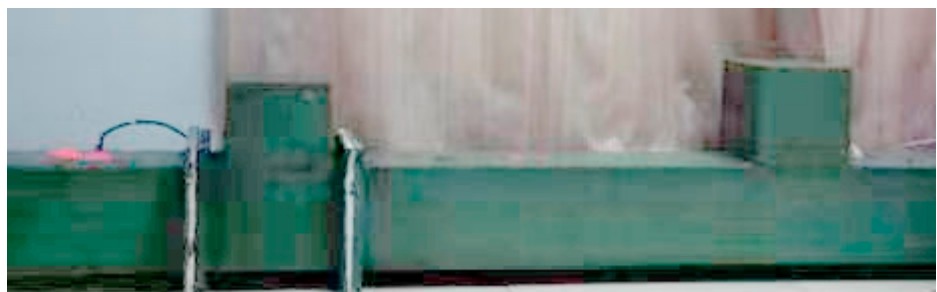

**Figure 4.** Absorption duct acoustics.

The impulse response information of the primary path and the secondary path are acquired by using the offline modeling method, as shown in Figure 5a,b. The total acoustic delay is estimated to be the secondary path impulse response, as shown in Figure 6.

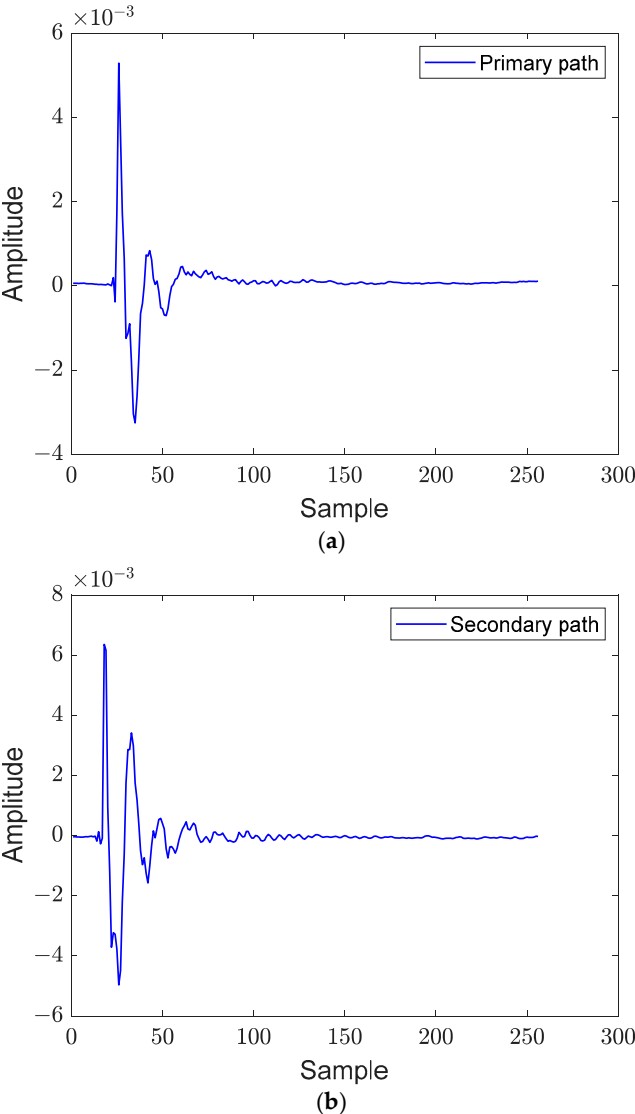

**Figure 5.** Impulse responses (**a**) primary path; (**b**) secondary path.

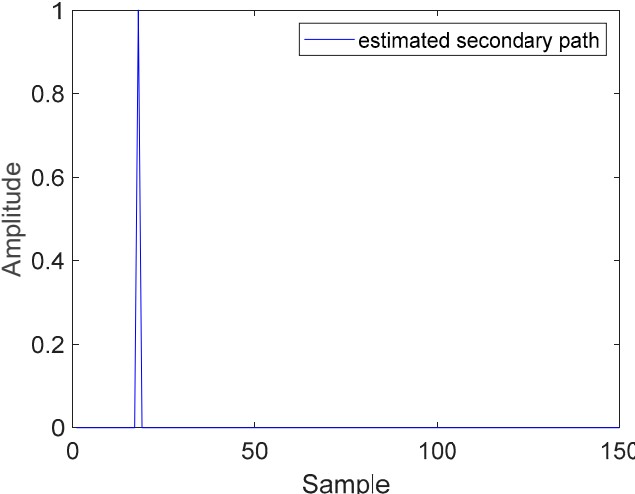

**Figure 6.** Total delay of estimated secondary path.

To ensure the stability of the narrowband feedback active noise control system, the optimum step size is chosen using repeated tests to achieve the fastest convergence rate

for the SFDFBANC algorithm and the other two algorithms. In the following simulation, the primary noise sources are single-frequency noise and narrowband noise, including narrowband white noise and narrowband pink noise. The stability characters, convergence speed, and noise reduction are also compared and analyzed among the three algorithms in the simulation.

### 3.1. Single-Frequency Simulation Result Analysis

In this simulation, the system sampling frequency was 2400 Hz, the primary noise was a single-frequency noise of 320 Hz, and the adaptive filter length was set at 256. The proposed SFDFBANC algorithm's step size was set at 0.4, the WSMANC algorithm's step size was 0.7, and the SAIMC algorithm's step size was 0.8.

Figure 7 shows the time-domain error signal curve of the SFDFBANC algorithm. It began to converge at about 7400 sampling numbers, taking about 3 s, and gradually stabilized at about 100,000 sampling points. The error signal spectrums after and before control of the three algorithms are shown and compared in Figure 8. When the three algorithms have achieved a steady state, the SFDFBANC algorithm has similar noise reduction to that of both the SAIMC algorithm and SWMANC algorithm. After control, the noise reduction achieved was 19.8 dB.

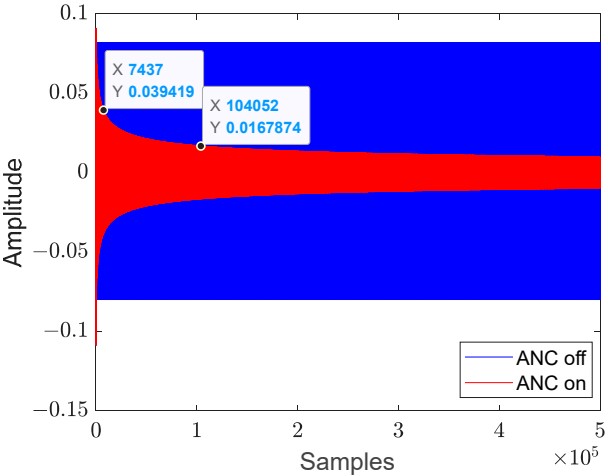

**Figure 7.** SFDFBANC algorithm time-domain error signal.

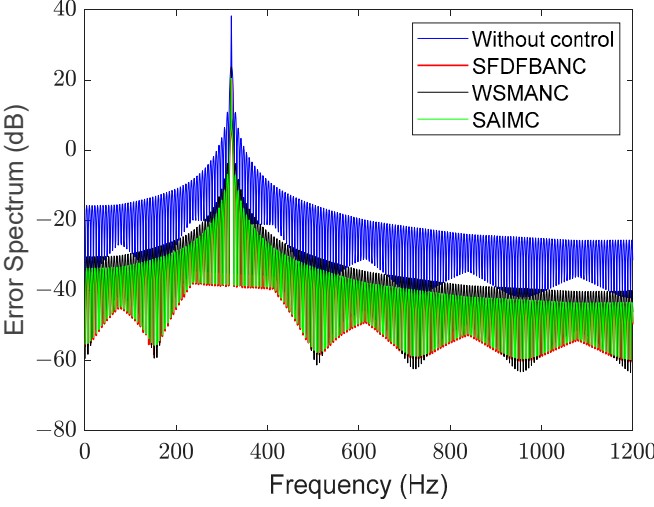

**Figure 8.** The three algorithms' error signal spectrum.

*3.2. Narrowband Noise Simulation Result Analysis*

3.2.1. Narrowband White Noise Simulation Result Analysis

In this simulation, the noise source was narrowband white noise with a frequency range of [300–330] Hz. The system sampling frequency was 2400 Hz. First, a bandpass filter with a length of 256 was designed; its frequency range is between 300 Hz and 330 Hz. Narrowband white noise is obtained by passing white noise through the designed bandpass filter.

In the simulation experiment, a step size of 0.04 was chosen in the proposed SFDF-BANC algorithm. The WSMANC algorithm's step size was 0.05, and the SAIMC algorithm's step size was 0.08. The MSE curve is compared among the three algorithms, which refers to the least mean square error, which can be seen in Figure 9. Compared to the SAIMC algorithm, the WSMANC algorithm's structure was simplified because the secondary path was replaced by direct acoustic delay; however, the two algorithms were realized by the principle of the time-domain filtered-$x$ LMS algorithm. Therefore, both the WSMANC algorithm and the SAIMC algorithm have a similar convergence character. Compared to these algorithms, the SFDFBANC algorithm has the fastest convergence rate, which is obtained by utilizing the principle of the block LMS algorithm method.

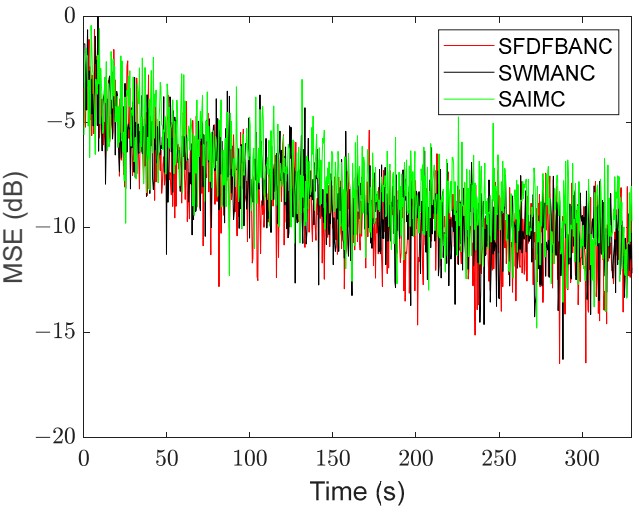

**Figure 9.** MSE curves of the three algorithms.

The noise reduction performances were also compared among the three algorithms. Before and after the control, the three algorithms' error signal spectra are shown in Figure 10. A steady state was reached, and the noise reduction effect of the SFDFBANC algorithm was equivalent to that of the WSMANC algorithm and the SAIMC algorithm. It was about 13.4 dB compared to the previous control.

3.2.2. Narrowband Pink Noise Simulation Result Analysis

Pink and white noise are common noises in everyday life. The spectra of the two types of noise are shown in Figure 11. Pink noise is mainly concentrated at a low frequency, such as fan noise, noise in the car or train, etc. ANC technology is much more effective at reducing low-frequency noise.

For the narrowband feedback control system, the sampling frequency was 2400 Hz, and the noise source was narrowband pink noise. It has a frequency range of 300 Hz to 330 Hz in this simulation. First, pink noise was generated using the audio software Cool Editor Pro 2.1; narrowband pink noise was obtained using pink noise filtered through the designed bandpass filter with a length of 256 and a frequency range of [300–330] Hz.

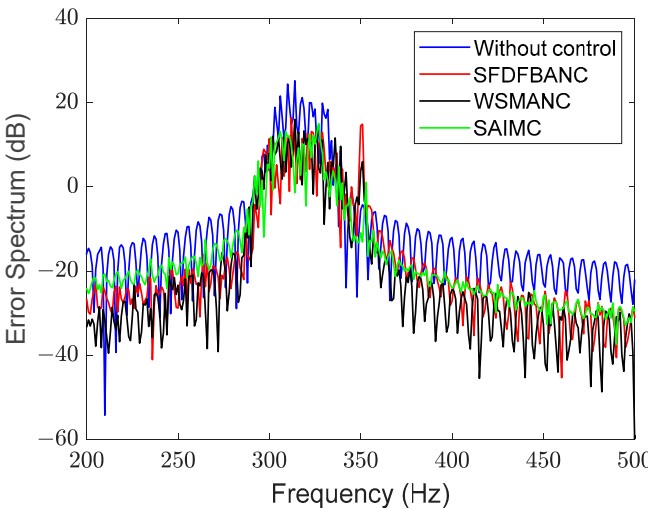

**Figure 10.** The three algorithms' error signal spectrum.

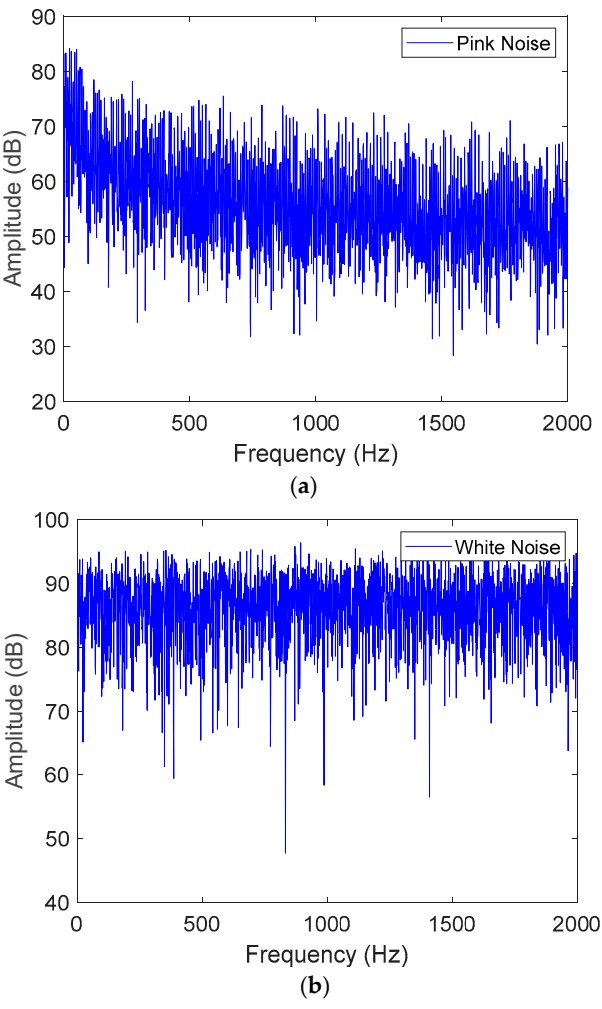

**Figure 11.** Spectra of two different noises (**a**) pink noise; (**b**) white noise.

In the simulation, the proposed SFDFBANC algorithm had a step size of 0.5, the WSMANC algorithm's step size was 0.6, and the step value of the SAIMC algorithm was set at 0.7. The MSE curves of the three algorithms are shown in Figure 12. Among the three algorithms, both the WSMANC algorithm and the SAIMC algorithm have equivalent

convergence speeds; however, the SFDFBANC algorithm has the fastest convergence rate due to adopting the structure of the block LMS algorithm.

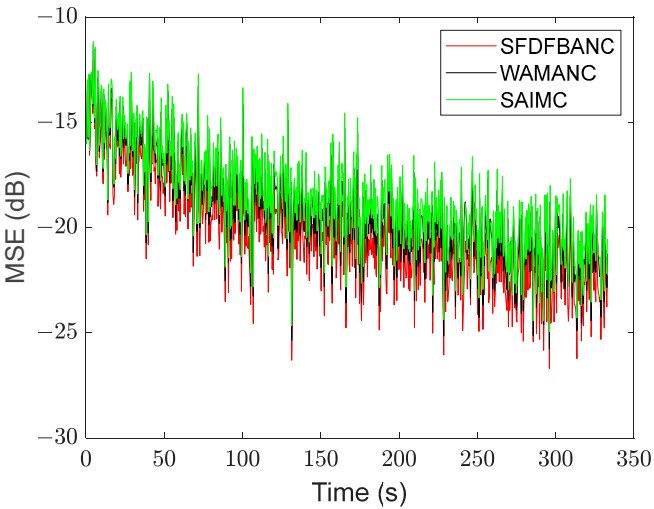

**Figure 12.** MSE curves of the three algorithms.

The comparison diagram of the three algorithms' error signal spectra after and before the control is shown in Figure 13. The noise reduction effect of the SFDFBANC algorithm was equivalent to that of both the WSMANC and SAIMC algorithms. After control, the noise reduction was about 12.5 dB.

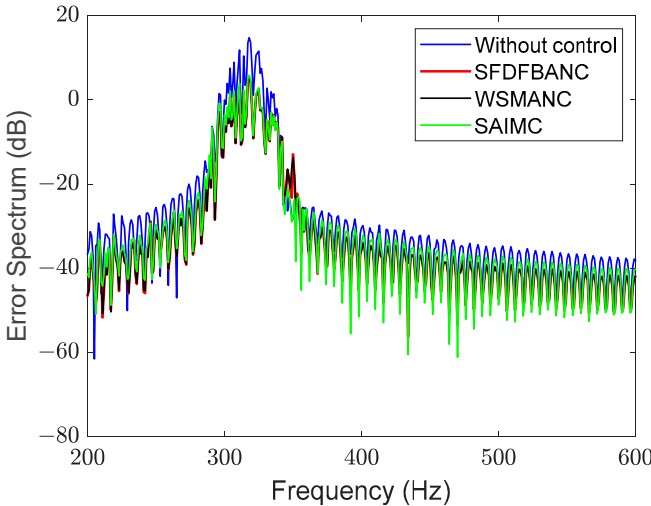

**Figure 13.** The three algorithms' error signal spectrum.

### 3.3. Fan Noise Simulation Result Analysis

The frequency domain feedback ANC algorithm (FDFBANC for simple) is frequently used, which is based on the theory of internal model control. It has a simple structure and good steady state character. In contrast with the proposed SFDFBANC algorithm, the secondary path impulse response needs to be estimated for the FDFBANC algorithm.

In this simulation, fan noise was adopted as the primary noise source; its frequency spectrum is shown in Figure 14. It can be filtered using a band pass filter to obtain a narrowband noise with a frequency range between 320 Hz and 350 Hz. The steady-state and noise-reduction properties of the proposed SFDFBANC and FDFBANC algorithms were compared in this simulation. The proposed SFDFBANC algorithm's step size was set at 0.7, and the FDFBANC algorithm's step size was 0.9. The acoustic environment adopted a sound absorption duct sound field, as shown in Figure 4. Figure 5's impulse response

was used for the FDFBANC algorithm in this simulation. Both of the algorithms' error spectra were shown in Figure 15. It can be seen that the proposed SFDFBANC algorithm has a similar noise reduction to that of the FDFBANC algorithm.

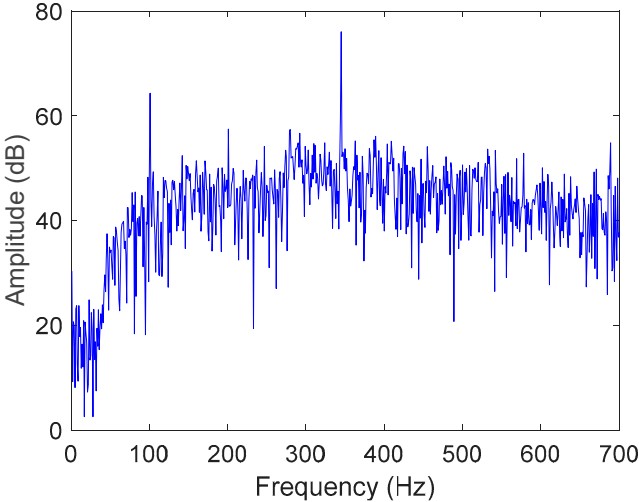

**Figure 14.** Spectrum of fan noise.

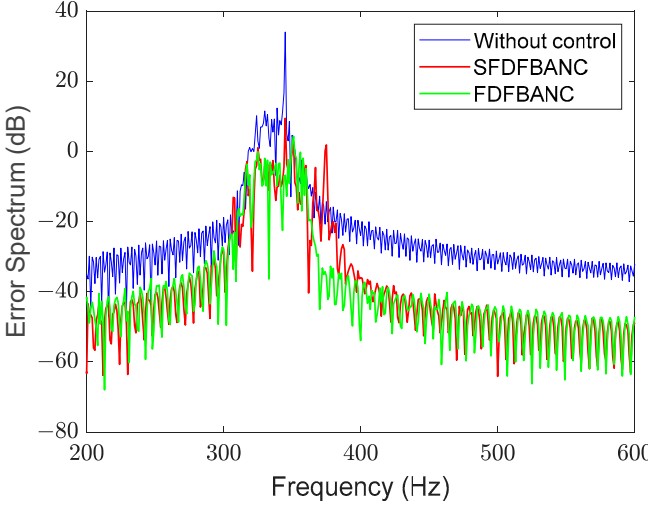

**Figure 15.** Error signal spectra of SFDFBANC and FDFBANC algorithms.

Compared to before the ANC control, the proposed SFDFBANC algorithm's noise reduction is about 14 dB.

Both of the algorithms' minimum mean square error (MSE) curves are shown in Figure 16. It can be seen that both of the algorithms have comparable convergence rates; however, the steady-state error of the SFDFBANC algorithm is lower than that of the FDFBANC algorithm before 150 s. For the FDFBANC algorithm, the perfect model was adopted, so the internal synthetic signal $x(n)$ is highly correlated with expected signal $d(n)$. While the proposed SFDFBANC algorithm directly used the error signal $e(n)$ as the internal reference signal $x(n)$, it has a lower correlation with the expected signal than the FDFBANC algorithm. As the time grows, the proposed SFDFBANC algorithm gradually closes the gap with the FDFBANC algorithm due to the feedback of the error signal $e(n)$. After 200 s, the proposed SFDFBANC algorithm was almost consistent with the MSE of the FDFBANC algorithm.

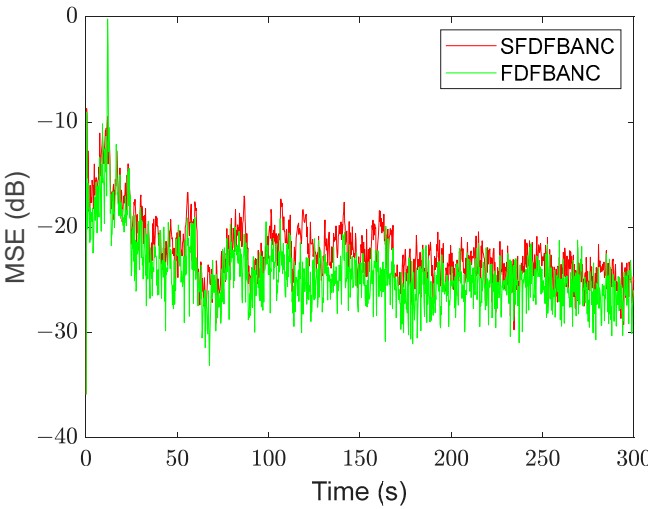

**Figure 16.** MSE of the two algorithms.

### 3.4. The Influence of Step Size Analysis

In this simulation, the primary noise was the narrowband white noise with a frequency range between 320 Hz and 350 Hz. The influence of step size $\mu$ was analyzed. It mainly affected the convergence rate and the steady state error of the SFDFBANC algorithm. The two different step sizes $\mu$ were chosen to realize the proposed SFDFBANC algorithm. One step size $\mu$ was 0.04 and the other was 0.05. Figure 17 shows the time domain error signal curves with different step sizes. The convergence rate responded to a step size of 0.05 faster than it did to a step size of 0.04, but the steady state error was larger for a step size of 0.05 than it was for a step size of 0.04. In Figure 17, when the step size $\mu$ was 0.05, the proposed SFDFBANC algorithm was unstable in nearly $7.5 \times 10^5$ samples. However, when the step size $\mu$ was 0.04, the SFDFBANC algorithm had better steady-state characteristics than it did when the step size was 0.05.

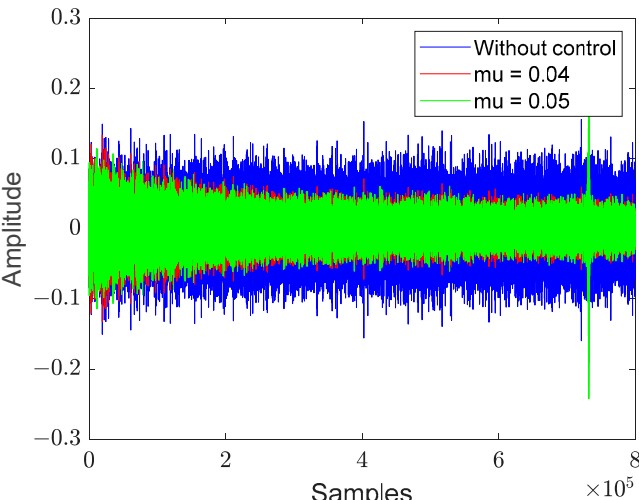

**Figure 17.** Time-domain error signal curves with large step size.

In this simulation, the time domain error signal curves between step sizes 0.04 and 0.03 were compared. The time domain error signal curves are shown in Figure 18. The convergence rate of step size 0.04 is faster than that of step size 0.03.

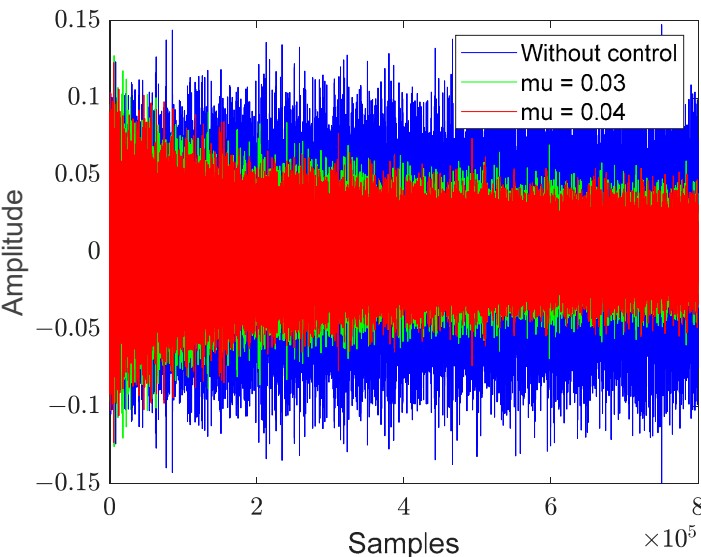

**Figure 18.** Time-domain error signal curves with small step size.

Due to the set of simulations, if the larger step size $\mu$ is chosen, the convergence rate will be accelerated; however, the steady state error will be increased. If the step size $\mu$ was set to be too large, the system may be unstable. If the smaller step size $\mu$ is chosen, the steady state error will be reduced, and the convergence rate will be slowed down. The optimal step size was the maximum step size without divergence of the SFDFBANC algorithm by trial and error.

## 4. Computational Complexity

The computation complexity of the three algorithms is analyzed in this section. Firstly, the relationship was studied between the real number multiplication computation and the variation of the adaptive filter length $N$. Secondly, the computational complexity of the three algorithms at different lengths $N$ was compared.

The proposed SFDFBANC algorithm's computational complexity mainly consists of five parts: (1) producing the output signal of the adaptive filter; (2) estimating of the total delay of the secondary path; (3) generating the frequency domain filtered reference signal; (4) generating the frequency domain error signal; and (5) updating the frequency domain adaptive filter coefficients. The computation of the WSMANC algorithm is divided into four parts: (1) controller output; (2) estimating total delay; (3) generating the filtering reference signal; and (4) updating the time domain adaptive filter coefficients. The computation of the SAIMC algorithm is divided into four parts: (1) controller output; (2) acquiring the secondary path information using the offline modeling method; (3) generating the filter reference signal; and (4) updating the time domain adaptive filter coefficients. The computation of the identical step (i.e., Step (1)) in the three algorithms was ignored, and the computation is calculated using $N$ sampling points.

For the three algorithms, the relationships between the computational complexity and the adaptive filter length $N$ were shown in Table 1. The calculation of the three algorithms is calculated by using different lengths of the adaptive filter $N$. This can be seen in Table 2.

When $N$ was 16, the SAIMC algorithm's computation complexity is 1040 real number multiplications, that of the WSMANC algorithm was 544 real number multiplications, and the proposed SFDFBANC algorithm's computation complexity was 736 real number multiplications. At that time, the SAIMC algorithm's calculation was higher than the SFDFBANC algorithm's calculation. When $N$ was 64, the calculation complexity of the SAIMC algorithm was 16,448 real number multiplications, and the WSMANC algorithm's calculation was 5248 real number multiplications; however, 3712 real number multiplications were calculated in the proposed SFDFBANC algorithm. In that case, both the

WSMANC and SAIMC algorithms had higher calculation complexities than that of the SFDFBANC algorithm.

**Table 1.** Relationships between the calculation complexity and the adaptive filter length $N$.

| Three Algorithms | Secondary Path Modeling | Time-Domain Filter Reference Signal | Frequency-Domain Filter Reference Signal | Frequency-Domain Error Signal | Frequency-Domain Filter Coefficient Update | Time-Domain Filter Coefficient Update | Computational Complexity |
|---|---|---|---|---|---|---|---|
| FDSFBANC | 0 | 0 | $4N\log_2 2N + 8N$ | $2N\log_2 2N$ | $8N$ | 0 | $6N\log_2 2N + 16N$ |
| WSMANC | 0 | $N\Delta$ | 0 | 0 | 0 | $N^2$ | $N^2 + N\Delta$ |
| SAIMC | $2N^2 + N$ | $N^2$ | 0 | 0 | 0 | $N^2$ | $4N^2 + N$ |

**Table 2.** The computational complexity (real number multiplications) of the three algorithms at different lengths of the adaptive filter $N$.

| Adaptive Filter Length $N$ | SAIMC | WSMANC | FDSFBANC |
|---|---|---|---|
| 2 | 18 | 40 | 56 |
| 4 | 100 | 88 | 136 |
| 8 | 264 | 208 | 320 |
| 16 | 1040 | 544 | 736 |
| 32 | 4128 | 1600 | 1664 |
| 64 | 16,448 | 5248 | 3712 |
| 128 | 65,664 | 18,688 | 8192 |
| 256 | 262,400 | 70,144 | 17,920 |
| 512 | 1,049,088 | 271,360 | 38,912 |

When $N$ was 256, there were 262,400 real number multiplications in the SAIMC algorithm, and the WSMANC algorithm's calculation was 70,144 real number multiplications; however, there were only 17,920 real number multiplications in the SFDFBANC algorithm. At that time, the WSMANC algorithm's calculation was more than four times that of the SFDFBANC algorithm, and the SAIMC algorithm's calculation was more than 15 times that of the SFDFBANC algorithm. When $N$ was 512, the WSMANC algorithm's calculation complexity was about 12.7 times that of the proposed SFDFBANC algorithm, and that of SAIMC algorithm was about 50 times that of the SFDFBANC algorithm's calculation.

As is shown in Tables 1 and 2, the three algorithms' calculation are increased by increasing the length value $N$. The calculation complexities of the WSMANC and SAIMC algorithms are related to $N^2$, which increases in the form of a power function ($N = 2^M$) and grows faster, while the relationship between the computational complexity and the adaptive length $N$ is a logarithmic relationship in the SFDFBANC algorithm, which grows more slowly than the power function. If the adaptive controller length $N$ is not less than 64, the SFDFBANC algorithm exhibits the least calculation. The change relationship between the calculation and the adaptive filter length $N$ ($N = 2^M$, $M$ is a natural number) is shown in Figure 19.

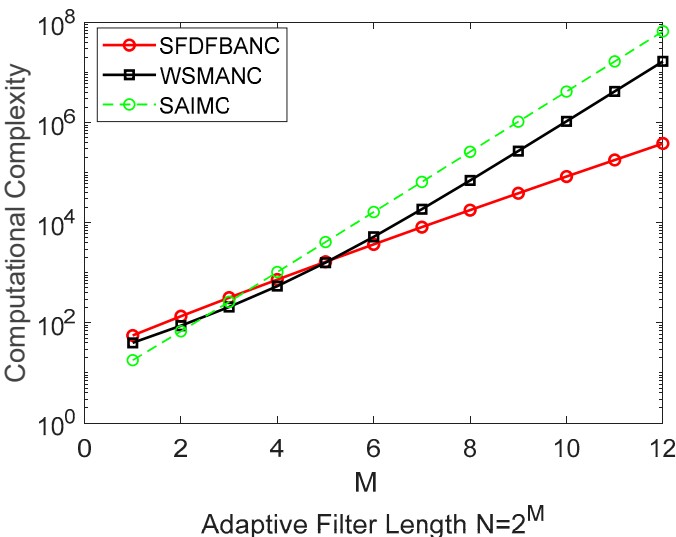

**Figure 19.** Computational complexity.

## 5. Conclusions

Through the analysis of the WSMANC algorithm, the time-domain adaptive filter coefficient update process and the convolution calculation process of obtaining the time-domain filter reference signal led to an increase in the calculation of the WSMANC algorithm in the form of a power function. Aiming at decreasing the WSMANC algorithm's calculation, the SFDFBANC algorithm is proposed in the paper. The algorithm uses total acoustic delay as the estimated secondary path information, which can avoid the computational complexity of obtaining the secondary path information process by using the offline modeling or online modeling methods. It also implements the adaptive filter coefficient update process in the frequency domain and replaces the convolution computational complexity with the frequency-domain multiple to produce the filtered internal synthesis signal. As such, the computational complexity changes in the form of a logarithmic function by increasing the number *N* of the adaptive filter length, which reduces the calculation complexity. Compared to both the WSMANC and SAIMC algorithms, if only the variable *N* is not less than 64, the SFDFBANC algorithm has the least calculation complexity. The primary sound sources were single-frequency noise, narrowband white noise, and narrowband pink noise to verify the characters of the SFDFBANC algorithm. The proposed SFDFBANC algorithm has the fastest convergence rate, and the effect of noise reduction is equivalent to that of the WSMANC algorithm and the SAIMC algorithm after reaching the steady state.

In this paper, the signal frequency and narrowband noise signals were adopted as the noise sources. In forthcoming research, the main areas of study are as follows:

(1) The proposed algorithm will adopt the subband adaptive filter method to expand the primary noise frequency range. The proposed algorithm will be investigated for its potential application in broadband active noise control.

(2) Study the stability of the feedback ANC algorithm. There is a closed loop in the feedback ANC system, so the poles always exist in the system function. When the frequency of primary noise is located on the pole, the ANC system will be unstable. In the following study, we will research the adaptive method to obtain the frequency information of the pole, which can improve the stability of feedback ANC algorithm and help the proposed SFDFBANC algorithm in broadband active noise control applications.

**Author Contributions:** Conceptualization, Y.G., M.G. and G.Y.; methodology, M.G. and G.Y.; software, Y.G. and M.G.; validation, Y.G., M.G. and G.Y.; formal analysis, Y.G. and M.G.; investigation, Y.G., M.G. and G.Y.; resources, G.Y.; data curation, Y.G., M.G. and G.Y.; writing—original draft preparation, Y.G. and M.G.; writing—review and editing, G.Y.; visualization, Y.G. and G.Y.; supervision, G.Y.; project administration, Y.G.; funding acquisition, M.G. All authors have read and agreed to the published version of the manuscript.

**Funding:** This work was jointly supported by the National Natural Science Foundation of China under Grant (No. 51902300) and the Natural Science Foundation of Zhejiang Province under Grant (No. LQ19A040003).

**Institutional Review Board Statement:** Not applicable.

**Informed Consent Statement:** Not applicable.

**Data Availability Statement:** The original contributions presented in the study are included in the article, further inquiries can be directed to the corresponding author.

**Conflicts of Interest:** The authors declare no conflicts of interest. We declare that we do not have any commercial or associative interest that represents a conflict of interest in connection with the work submitted.

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
