# Peer review of "A Simplified Frequency-Domain Feedback Active Noise Control Algorithm"

_applsci, doi:10.3390/app14073084_

Round 1

Reviewer 1 Report

Comments and Suggestions for Authors

This paper, "A Simplified Frequency-Domain Feedback Active Noise Control Algorithm" addresses the computational challenges in active noise control (ANC) systems, especially regarding the complexity that arises with increasing adaptive filter length. The authors propose a simplified frequency-domain feedback ANC algorithm aimed at reducing computational complexity while maintaining noise reduction performance.

While the proposed SFDFBANC algorithm's reduction in computational complexity is clear, the paper could benefit from a more detailed comparison of its algorithmic innovations against the state-of-the-art. Highlighting specific novel aspects or modifications to existing frequency-domain ANC approaches can help underline the unique contributions of this work.

The simulations presented focus on specific types of noise (single-frequency, narrowband white, and narrowband pink noise). Expanding the discussion on these simulations to include broadband noise or real-world environmental noise could provide a more comprehensive evaluation of the algorithm's performance.

The paper mentions the selection of optimal step sizes for different algorithms but does not elaborate on how these were chosen or their impact on algorithm performance. Expand this statement, maybe Including a discussion or analysis of parameter sensitivity and selection criteria.

While simulations are valuable, testing the algorithm with real-world data or in practical ANC applications could significantly enhance the paper's relevance and demonstrate its effectiveness in actual noisy environments. Or provide literature review on the possible approaches.

In conclusion, a detailed discussion on potential adaptations, challenges, and future research directions for applying the algorithm to a broader spectrum of noise control scenarios would be beneficial.

Comments on the Quality of English Language

Acceptable

Author Response

Thank you for youe comments and suggestions.

please see the attachment, thank you.

Reviewer 2 Report

Comments and Suggestions for Authors

The authors propose the SFDFBANC algorithm. The algorithm uses the sum of the acoustic delay as an estimate of the secondary path, which avoids complex calculations. The work is interesting, well-written and, above all, developmental because, as the authors mention, in the upcoming research, an algorithm will be proposed for research in terms of its potential application in broadband active noise control

Author Response

Thank you for your comments and suggestions.

please see the attachment, thank you.

Reviewer 3 Report

Comments and Suggestions for Authors

The article presents an active noise control algorithm developed to reduce computational complexity especially when using adaptive filters with long filter lengths. The authors demonstrate that the performance of the algorithm is similar or better to other presented methods while requiring less computation.

My questions and remarks are:

1) Providing a detailed block diagram indicating the placement of the speakers, microphones, other electronics, and instruments would enhance the understanding of the experimental setup.

2) Please explain the rationale for downsampling the signal to 2400 Hz. Please discuss the potential limitations this low sampling rate might introduce for specific applications.

3) In Figure 1. and Figure 2. x(n) and e(n) are connected to each other by a line. Please clarify the function of this element and its role in the signal path.

4) Figure 3 would be more informative as a block diagram illustrating the position of the hardware elements.

5) The bandwidth of the “Narrowband Pink Noise” is much less than a decade. The limited bandwidth of the 'Narrowband Pink Noise' seems to offer minimal deviation from a band-limited white noise. What is the purpose of including this specific noise type.

The approach to use FFT to reduce computational cost is promising. In my opinion the work can be suitable for publication after addressing the beforementioned issues.

Author Response

(The authors gave the same response as above.)

Round 2

Reviewer 1 Report

Comments and Suggestions for Authors

The comments are properly addressed, I now suggest the manuscript can be accepted in this current form,

Comments on the Quality of English Language

The comments are properly addressed, I now suggest the manuscript can be accepted in this current form,